# LANGUAGE MODELS ARE MULTILINGUAL CHAIN-OF-THOUGHT REASONERS

**Freda Shi**[1,2,∗]    **Mirac Suzgun**[1,3,∗]    **Markus Freitag**[1]    **Xuezhi Wang**[1]

**Suraj Srivats**[4]    **Soroush Vosoughi**[4]    **Hyung Won Chung**[1]    **Yi Tay**[1]

**Sebastian Ruder**[1]    **Denny Zhou**[1]    **Dipanjan Das**[1]    **Jason Wei**[1]

[1]Google Research    [2]Toyota Technological Institute at Chicago
[3]Stanford University    [4]Dartmouth College

## ABSTRACT

We evaluate the reasoning abilities of large language models in multilingual settings. We introduce the Multilingual Grade School Math (MGSM) benchmark, by manually translating 250 grade-school math problems from the GSM8K dataset (Cobbe et al., 2021) into *ten* typologically diverse languages. We find that the ability to solve MGSM problems via chain-of-thought prompting emerges with increasing model scale, and that models have strikingly strong multilingual reasoning abilities, even in underrepresented languages such as Bengali and Swahili. Finally, we show that the multilingual reasoning abilities of language models extend to other tasks such as commonsense reasoning and word-in-context semantic judgment. The MGSM benchmark is publicly available at https://github.com/google-research/url-nlp.

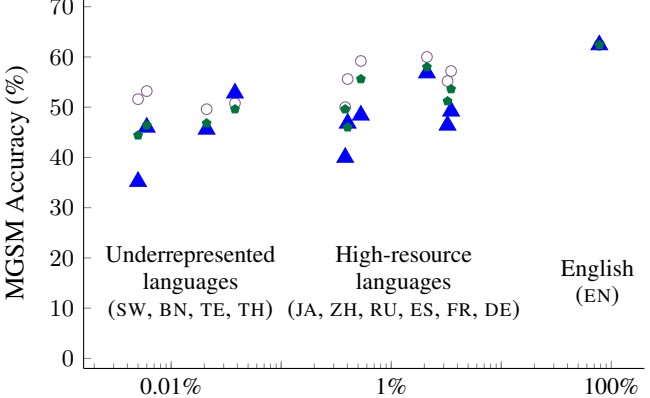

Figure 1: Correlation between language frequency and MGSM accuracy for PaLM-540B. The accuracy is surprisingly high, even for underrepresented languages like Swahili (SW) and Bengali (BN), which account for less than 0.01% of the pre-training dataset.

---

∗Equal contribution. Work done during internship at Google Research.

# 1 INTRODUCTION

Recent work has shown that presenting explicit reasoning steps (i.e., chains of thought; CoT) in English elicits multi-step reasoning abilities of large language models such as GPT-3 and PaLM (Brown et al., 2020; Chowdhery et al., 2022; Wei et al., 2022b, *inter alia*). Pretrained multilingual language models have also achieved impressive performance on various NLP tasks across typologically distinct languages (Conneau et al., 2020; Xue et al., 2021; Chowdhery et al., 2022; Clark et al., 2020; Hu et al., 2020; Ruder et al., 2021, *inter alia*). Tasks in existing multilingual benchmarks usually require only simple reasoning steps, and so it is still unclear how well language models perform on tasks that require more complex reasoning in a multilingual setting.

In this work, we introduce the **MGSM** benchmark to bridge the gap between the progress on English-based chain-of-thought reasoning and multilingual NLP. We extend a subset of the English-language GSM8K dataset (Cobbe et al., 2021) to ten typologically diverse languages via manual translation of problems into target languages. To the best of our knowledge, this is the first multilingual benchmark to evaluate the arithmetic reasoning abilities of language models.

We evaluate two large language models, GPT-3 (Brown et al., 2020; Ouyang et al., 2022) and PaLM (Chowdhery et al., 2022), on this benchmark. While both models solve less than 20% of problems with standard prompting, the 540-billion-parameter PaLM model in particular shows exceptional multilingual reasoning abilities with intermediate reasoning steps (Figure 1), solving more than 40% of the problems in any investigated language, including underrepresented languages such as Bengali and Swahili. In our best setting, PaLM achieves an average solve rate of 55% across languages. We find that intermediate reasoning steps in English consistently lead to competitive or better results than those written in the native language of the question, suggesting that English chain-of-thought prompting may be a useful baseline for future multilingual reasoning work.

We further demonstrate that the multilingual reasoning abilities of pretrained models extend to common-sense reasoning (Ponti et al., 2020) and word-in-context semantic judgment (Raganato et al., 2020). By presenting the models with few-shot examples in different languages, PaLM sets a new state-of-the-art performance (89.9%) on XCOPA (Ponti et al., 2020), outperforming the prior approaches that require thousands of training examples.

# 2 THE MGSM BENCHMARK

In this section, we describe the collection process of Multilingual Grade School Math (MGSM), to our knowledge the first multilingual arithmetic reasoning benchmark.

**Source data.** We used GSM8K (Cobbe et al., 2021), an English-language human-annotated grade-school math problem dataset, as the base data source. For MGSM, we took the first 250 examples from the GSM8K official test example list. Each problem requires two to eight steps to solve according to the official solution (Figure 2). The answer for each question in GSM8K was written as an Arabic numeral, which we kept consistent across all languages to facilitate cross-lingual prediction.[1]

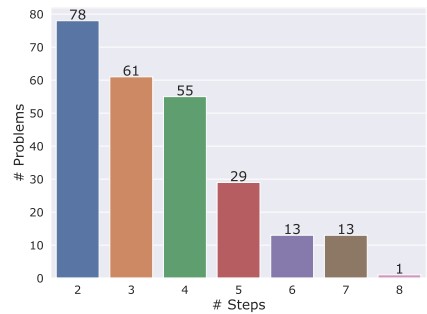

Figure 2: MGSM problem distribution with respect to the number of reasoning steps in the standard solution.

**Target language selection.** We selected a typologically diverse set of ten languages other than English (EN), spanning eight language families and different levels of representation in standard pretraining datasets such as mC4 (Xue et al., 2021): Bengali (BN), Chinese (ZH), French (FR), German (DE), Japanese (JA), Russian (RU), Spanish (ES), Swahili (SW), Telugu (TE), and Thai (TH).

---

[1]Certain scripts such as Devanagari employ different numerals. We restrict the data to Arabic numerals for consistency but future work may investigate cross-lingual numeracy by mapping Arabic numerals to those of the corresponding script (see Spithourakis & Riedel, 2018).

| Original Question | Frage: Roger hat 5 Tennisbälle. Er kauft noch 2 Dosen Tennisbälle. In jeder Dose sind 3 Tennisbälle. Wie viele Tennisbälle hat er jetzt? |
|---|---|
| DIRECT | Antwort: 11 |
| NATIVE-COT | Schritt-für-Schritt-Antwort: Roger begann mit 5 Bällen. 2 Dosen von jeweils 3 Tennisbällen macht 6 Tennisbälle. 5 + 6 = 11. Die Antwort ist 11. |
| EN-COT | Step-by-Step Answer: Roger started with 5 balls. 2 cans of 3 tennis balls each is 6 tennis balls. 5 + 6 = 11. The answer is 11. |
| **Translated English Question** | Question: Roger has 5 tennis balls. He buys 2 more cans of tennis balls. Each can has 3 tennis balls. How many tennis balls does he have now? |
| TRANSLATE-EN | Step-by-Step Answer: Roger started with 5 balls. 2 cans of 3 tennis balls each is 6 tennis balls. 5 + 6 = 11. The answer is 11. |

Table 1: Example solution formats (§3) for a German exemplar problem, where German-specific components are underlined and are changed to the corresponding translations for other investigated languages. For DIRECT, NATIVE-COT and EN-COT, we provide the original German question as input to the model and expect an answer in the corresponding format; for TRANSLATE-EN, we input the translated question in English, and expect a step-by-step solution in English. To obtain the desirable output format, we prepend few-shot examples in the corresponding format.

**Manual translation process.** We enlisted the help of paid professional translators (two for Chinese and German, three for Russian, five for Thai, one for each remaining target language) for the manual translation of the 250 selected English-language examples from GSM8K, through professional translation services.[2] All translators involved were native speakers of the target language and had at least two years of professional experience in translating between English and the target language. All translators had signed a machine translation (MT) non-usage declaration before they started to work. To verify the quality of the human translations, the vendor sent a random subset of translations to an additional translator to verify the quality, and checked for $n$-gram overlap with popular MT providers, such as Google Translation and Bing Microsoft Translator, to ensure that no machine translation toolkit has been used. We employ the translation results as gold standard translations.

## 3 MULTILINGUAL CHAIN-OF-THOUGHT PROMPTING

We provide an overview of standard prompting and chain-of-thought prompting, as well as their extensions to the multilingual setting, which we illustrate in Table 1 and use in our experiments (§4).

In standard prompting, given a prompt in the source language, the model is asked to predict the answer (Brown et al., 2020; Schick & Schütze, 2021). This can be done in a zero-shot or few-shot setting by providing exemplars following the same template as additional input to the model. We refer to this setting as **direct answer prediction (DIRECT)** as the model directly predicts the answer to the problem. This setting measures the model's ability to solve problems without any intermediate reasoning steps.

Chain-of-thought (COT; Wei et al., 2022b) prompting helps improve many few-shot reasoning tasks, by augmenting few-shot examples with intermediate reasoning steps that should be predicted by the model. In the multilingual setting, we can apply CoT to **solve the problem in the native language (NATIVE-COT)** by predicting the reasoning steps in the original language of the problem. This measures the model's ability to both understand and solve the problem in a specific language.

Alternatively, we can ask the model to **predict the chain of thought in English (EN-COT)**, regardless of the problem language. Such an approach may be useful as English is often used as the source language for cross-lingual transfer (Hu et al., 2020) and has been found effective when used as the prompt language (Zhao & Schütze, 2021; Winata et al., 2021; Lin et al., 2021b).

In addition, Kojima et al. (2022) find that **zero-shot CoT** is surprisingly effective on reasoning tasks, by prompting the model with "*Let's think step by step.*" without any exemplar. They then concatenate

---

[2]https://www.vengaglobal.com/technology/translation-assets

| | DIRECT | NATIVE-CoT | EN-CoT | TRANSLATE-EN |
|---|---|---|---|---|
| NATIVE-EXEMPLARS | ✓ | ✓ | ✓ | ✓ |
| ENGLISH-EXEMPLARS | ✓ | N/A | ✓ | N/A |
| MULTILINGUAL-EXEMPLARS | ✓ | ✓ | ✓ | N/A |

Table 2: Possible combinations between few-shot exemplar selection and solution strategies.

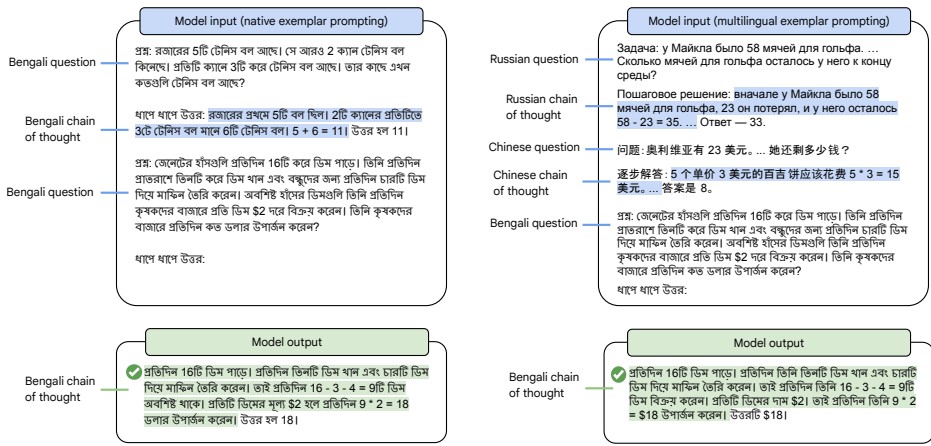

Figure 3: The chain-of-thought prompts and example model outputs in the MGSM experiments. The solutions are written in the same language as the questions of interest (NATIVE-CoT).

the question, model response and "*The answer is*" to prompt the model again for the final answer. In this work, we evaluate zero-shot CoT by solving both in English (**EN-CoT-0SHOT**) which uses the prompting sentences above, and in the native language of the problem (**NATIVE-CoT-0SHOT**) which uses the translated prompting sentences in the corresponding languages with Google Translate.

Finally, we can **translate the problem to English and solve it with English CoT (TRANSLATE-EN)**. In this setting, we use the Google Translate API to translate problems into English. This mirrors the translate-train setup (Hu et al., 2020; Xue et al., 2021; Ruder et al., 2021), the best-performing setting for fine-tuning multilingual models where the training data is translated to English.

Beyond the prompting methods, there are different ways to provide few-shot examples in context for multilingual prompting:

- **All native question exemplars (NATIVE-EXEMPLARS).** We use a few in-language questions together with their solutions as the few-shot prompt exemplars. This is the most natural setting when we have a few examples in each investigated language.

- **All English question exemplars (ENGLISH-EXEMPLARS).** When we are unable to access any existing questions or solution examples in some languages, an intuitive way is to use English questions and solutions as exemplars to perform zero-shot cross-lingual transfer. Note that it is unrealistic to combine this exemplar selection setting with NATIVE-CoT, since we assume no access to the native language for prompting.

- **Generic multilingual question exemplars (MULTILINGUAL-EXEMPLARS).** Similar to ENGLISH-EXEMPLARS, we assume access to questions and solutions in a few languages, and test if multilingual exemplars better elicit the multilingual reasoning ability of models.

For TRANSLATE-EN, as all exemplar questions and solutions are in English, we only experiment with the translated native question exemplars and English CoT. We summarize the combinations of prompting and exemplar methods in Table 2, and present an illustration in Figure 3. Detailed prompting input for each investigated combination can be found in Appendix A.2.

| | AVG | HRL | URL | EN | DE | FR | ES | RU | ZH | JA | TH | TE | BN | SW |
|---|---|---|---|---|---|---|---|---|---|---|---|---|---|---|
| Lang. Freq. (PaLM, %) | – | – | – | 78.0 | 3.5 | 3.3 | 2.1 | .53 | .40 | .38 | .04 | .02 | .006 | .005 |
| **GPT-3 (`text-davinci-002`)** | | | | | | | | | | | | | | |
| • Direct | 11.7 | 15.1 | 5.7 | 16.0 | 14.8 | 16.8 | 17.2 | 12.4 | 18.0 | 11.2 | 8.8 | 0.8 | 4.4 | 8.8 |
| • Native-CoT | 26.4 | 34.7 | 7.2 | 53.6 | 36.0 | 37.6 | 40.4 | 28.4 | 40.0 | 26.0 | 10.8 | 0.4 | 6.4 | 11.2 |
| • EN-CoT | 31.6 | 39.4 | 13.9 | 53.6 | 44.0 | 46.0 | 44.8 | 28.4 | 40.8 | 32.4 | 19.6 | 5.6 | 9.6 | 20.8 |
| • Native-CoT-0shot | 9.8 | 7.8 | 3.6 | 46.8 | 8.4 | 4.8 | 13.2 | 6.4 | 5.2 | 8.8 | 5.2 | 1.2 | 2.4 | 5.6 |
| • EN-CoT-0shot | 29.5 | 37.2 | 13.7 | 46.8 | 40.0 | 36.0 | 42.4 | 40.0 | 33.2 | 31.6 | 19.6 | 6.8 | 15.2 | 13.2 |
| • Translate-EN | 45.6 | 47.5 | 40.7 | 53.6 | 46.4 | 46.4 | 51.6 | 48.8 | 47.2 | 44.8 | 41.2 | 42.8 | 41.2 | 37.6 |
| **PaLM-540B** | | | | | | | | | | | | | | |
| • Direct | 18.6 | 19.3 | 16.8 | 22.0 | 18.8 | 19.6 | 20.0 | 22.0 | 19.2 | 16.0 | 16.8 | 17.6 | 17.2 | 15.6 |
| • Native-CoT | 48.1 | 47.9 | 44.9 | **62.4** | 49.2 | 46.4 | 56.8 | 48.4 | 46.8 | 40.0 | 52.8 | 45.6 | 46.0 | 35.2 |
| • EN-CoT | 51.3 | 52.3 | 46.8 | **62.4** | 53.6 | 51.2 | 58.0 | 55.6 | 46.0 | 49.6 | 49.6 | 46.8 | 46.4 | 44.4 |
| • Native-CoT-0shot | 14.4 | 13.2 | 7.7 | 48.0 | 12.8 | 12.4 | 16.8 | 13.6 | 10.8 | 12.8 | 7.6 | 6.8 | 6.8 | 9.6 |
| • EN-CoT-0shot | 30.8 | 38.3 | 15.2 | 48.0 | 38.4 | 36.0 | 42.4 | 42.0 | 35.6 | 35.2 | 20.0 | 10.4 | 14.0 | 16.4 |
| • Translate-EN | **55.0** | **56.3** | **51.2** | **62.4** | **57.2** | **55.2** | **60.0** | **59.6** | **55.6** | **50.0** | **50.8** | **49.6** | **53.2** | **51.2** |

Table 3: Accuracy (%) on MGSM of different models and languages with exemplar questions in native languages (Native-Exemplars). HRL: average performance across high-resource languages with larger than 0.1% frequency in the training corpora; URL: average performance across underrepresented languages. We use 6 questions and solutions as the few-shot exemplar whenever possible: while the token number for 6-shot prompts in some languages may exceed the token number limit of GPT-3, we use the maximum possible number of exemplars instead for these cases. Detailed numbers of exemplars for each language in GPT-3 experiments can be found in Appendix A.1. The best numbers in each column are in **boldface**.

# 4 Experiments on MGSM

In this section, we evaluate the multilingual reasoning abilities of two representative state-of-the-art pretrained large language models—GPT-3 and PaLM—on our MGSM benchmark in various prompting settings using exemplars in the source language (Native-Exemplars).[3] Throughout this paper, we generate outputs using greedy decoding (i.e., sampling with temperature $\tau = 0$).

## 4.1 Main Results

We first compare the few-shot Native-Exemplars performance with different solution strategies (Table 3). In line with the English results reported by Wei et al. (2022b), we find that intermediate reasoning steps (Native-CoT and EN-CoT) help both models achieve substantial reasoning performance gains across all languages, outperforming direct answer prediction with no explicit reasoning steps (Direct) by a significant margin. PaLM shows exceptional multilingual reasoning ability: while it outperforms GPT-3 on all languages with different settings, PaLM-540B with intermediate reasoning steps (Native-CoT and EN-CoT) achieves results similar to Translate-EN on all languages, even on underrepresented languages such as Bengali (BN) and Swahili (SW), which cover less than 0.01% of the training corpora.

In addition, reasoning in English (EN-CoT, EN-CoT-0shot) consistently achieves competitive or better performance than reasoning in the native language of the question (Native-CoT, Native-CoT-0shot), suggesting that English intermediate steps can be considered as useful baseline in future work on multilingual reasoning.

## 4.2 Further Analysis

**Effect of language frequency in training corpora.** We illustrate the main results of Native-CoT, EN-CoT and Translate-EN with respect to the language frequency in PaLM training data

---

[3] We focus on these two models due to their notable few-shot performance; in contrast, many other multilingual models are not as competitive as them in the same settings, and are generally used for finetuning with more data (Winata et al., 2021).

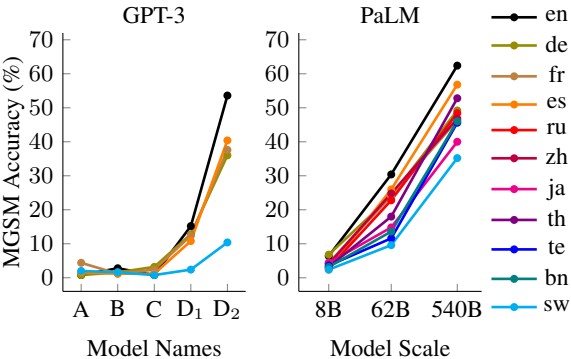
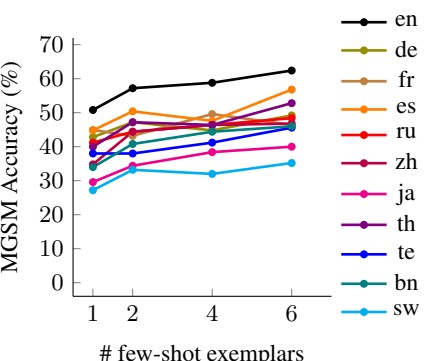

Figure 4: MGSM accuracy with different models. Letters A, B, C, D₁, and D₂ denote `text-ada-001`, `text-babbage-001`, `text-curie-001`, `text-davinci-001`, and `text-davinci-002` in the GPT-3 family, respectively. While the number of parameters in each GPT-3 model is not publicly available, we order them alphabetically. Detailed numbers can be found in Table 8.

Figure 5: MGSM accuracy of PaLM-540B with different numbers of few-shot exemplars. Detailed numbers can be found in Table 8.

(Figure 1). Surprisingly, there is no strong correlation between the performance and the language frequency in the training corpora: the average accuracy among the four underrepresented languages was only 3% lower than the that among the six high-resource languages (44.9% vs 47.9%). Moreover, the performance of reasoning in Thai, Telugu, and Bengali is on par with reasoning in French, Japanese, and Chinese, despite having significantly much less data in the training corpora.

In contrast to prior work that identifies language frequency as important for complex NLU tasks with relatively smaller models (Hu et al., 2020; Lauscher et al., 2020; Ahuja et al., 2022), these results thus indicate that the reasoning ability of large language models may not be primarily dependent on their presence in training data and that language models are able to transfer their knowledge from high-resource to underrepresented languages to some extent.

**Effect of model scale.** We analyze the effect of model scale (i.e., number of model parameters and computational resources used for training) on their multilingual arithmetic reasoning abilities (Figure 4). As the models scale up, the performance generally improves for both GPT-3 and PaLM model series on all languages. Neither model achieves a substantial solve rate until a certain scale (`text-davinci-001` for GPT-3 and PaLM-62B for PaLM), hence multilingual reasoning can be considered an *emergent ability* of large language models (Wei et al., 2022a). It is worth noting that the amount of training data per language is constant across language model scales for PaLM—the fact that scale facilitates reasoning implies that further scaling may continue to improve the multilingual reasoning ability of large language models.

**Effect of exemplar amount.** We analyze how the multilingual reasoning performance of PaLM-540B, the overall best-performing model, is affected by the number of few-shot exemplars (Figure 5). Although not all trends are strictly increasing with the number of exemplars, PaLM-540B benefits from having more examples in general for all languages.

**Effect of exemplar type choice.** We compare the multilingual reasoning performance of PaLM-540B across languages with different exemplar choices (Table 4). For the MULTILINGUAL-EXEMPLARS setting, we concatenate one example from each of the most frequent languages (English, German, French, Spanish, Russian, and Chinese) as the generic prompt for all languages. While the best choice is almost always to use NATIVE-EXEMPLARS and EN-COT, MULTILINGUAL-EXEMPLARS with EN-COT achieves competitive performance across the board, suggesting an effective approach when we do not have access to any existing example in some languages.

Most notably, with EN-COT, MULTILINGUAL-EXEMPLARS significantly outperforms ENGLISH-EXEMPLARS on all non-English languages, including those not covered by the few-shot examples,

| | AVG | HRL | URL | EN | DE | FR | ES | RU | ZH | JA | TH | TE | BN | SW |
|---|---|---|---|---|---|---|---|---|---|---|---|---|---|---|
| NATIVE-EXEMPLARS | | | | | | | | | | | | | | |
| NATIVE-CoT | 48.1 | 47.9 | 44.9 | **62.4** | 49.2 | 46.4 | 56.8 | 48.4 | 46.8 | 40.0 | 52.8 | 45.6 | 46.0 | 35.2 |
| EN-CoT | **51.3** | **52.3** | **46.8** | **62.4** | **53.6** | **51.2** | **58.0** | **55.6** | 46.0 | **49.6** | **49.6** | **46.8** | 46.4 | **44.4** |
| MULTILINGUAL-EXEMPLARS | | | | | | | | | | | | | | |
| NATIVE-CoT | 29.8 | 31.8 | 26.3 | 52.0 | 41.6 | 7.2 | 10.4 | 36.0 | 42.8 | 32.8 | 18.0 | 33.6 | 26.8 | 26.8 |
| EN-CoT | 48.7 | 50.0 | 46.3 | 57.6 | 53.2 | 43.2 | 53.2 | 48.0 | **51.2** | 43.6 | 46.8 | 46.4 | **48.4** | 43.6 |
| ENGLISH-EXEMPLARS | | | | | | | | | | | | | | |
| EN-CoT | 34.7 | 39.4 | 26.6 | **62.4** | 46.0 | 37.2 | 50.4 | 23.6 | 29.2 | 26.8 | 17.2 | 30.0 | 34.4 | 24.8 |

Table 4: Performance on MGSM with different prompt exemplar type choices: the first section is copied correspondingly from Table 3. The best numbers in each column are in **boldface**.

| MODEL | AVG | ET | HT | ID | IT | QU | SW | TA | TH | TR | VI | ZH |
|---|---|---|---|---|---|---|---|---|---|---|---|---|
| HUMAN | 97.6 | 98.2 | 96.4 | 100 | 97 | 94.8 | 99 | 98.6 | 98.2 | 96.4 | 98.4 | 96.6 |
| MAD-X Base | 61.0 | 61.3 | 53.7 | 65.8 | 63.0 | 52.5 | 56.3 | 61.9 | 61.8 | 60.3 | 66.1 | 67.6 |
| XLM-R Large | 68.7 | 71.4 | (50) | 79.8 | 72.6 | (50) | 59.2 | 73 | 72.8 | 74.4 | 73.8 | 78.6 |
| mT5-XXL | 74.9 | 77.5 | 72.1 | 81.1 | 75.9 | 54.5 | 74.1 | 75.9 | 78.3 | 78.1 | 76.9 | 79.5 |
| RoBERTa Large (TT) | 76.1 | 81.0 | 73.8 | 82.2 | 77.8 | (50) | 74.2 | 79.6 | 71.4 | 79.6 | 81.0 | 86.0 |
| Codex (`code-davinci-002`) | | | | | | | | | | | | |
| • DIRECT | 73.3 | 73.8 | 55.6 | 88.8 | 95.4 | 51.2 | 56.0 | 54.6 | 70.2 | 88.6 | 80.4 | 91.4 |
| • EN-CoT | 80.7 | 88.8 | 79.6 | 91.4 | 96.6 | 52.2 | 67.4 | 55.8 | 84.2 | 91.2 | 86.6 | 93.4 |
| PaLM-540B | | | | | | | | | | | | |
| • DIRECT | 83.7 | 77.4 | 78.0 | 92.6 | 96.0 | 61.0 | 69.4 | 85.4 | 87.2 | 92.8 | 89.8 | 91.6 |
| • EN-CoT | **89.9** | **91.0** | **89.6** | **94.0** | **97.4** | **66.8** | **85.4** | **90.8** | **90.2** | **94.6** | **94.6** | **94.8** |

Table 5: Accuracy on the XCOPA languages compared to previous work. Human evaluation (HUMAN) on XCOPA was performed by Ponti et al. (2020). The MAD-X Base, XLM-R Large, and RoBERTa Large (*translate test*) results are from Ponti et al. (2020), whereas the mT5 results are from (Ruder et al., 2021). Applying multilingual CoT-prompting to PaLM-540B has enabled us to achieve a new state-of-the-art performance on XCOPA. The best model result in each column is in **boldface**.

suggesting that a multilingual few-shot prompt helps elicit the multilingual reasoning abilities of models more effectively than a monolingual (English) one.

## 5 EXTENSION TO OTHER MULTILINGUAL REASONING BENCHMARKS

To better understand the multilingual reasoning abilities of large pretrained language models, we extend our experiments to two additional multilingual reasoning benchmarks, XCOPA (Ponti et al., 2020) and XL-WiC (Raganato et al., 2020). Throughout this section, we evaluate the Codex (`code-davinci-002`; Chen et al., 2021)[4] and PaLM-540B models.

### 5.1 XCOPA

XCOPA is a multilingual evaluation dataset designed to assess the causal commonsense reasoning capabilities of language models across multiple languages.[5] It is an extension and re-annotation of the English COPA dataset (Gordon et al., 2012) where the validation and test set examples are carefully translated to and annotated in 11 typologically diverse languages. These languages are Estonian (ET), Indonesian (ID), Italian (IT), Cusco-Collao Quechua (QU), Swahili (SW), Tamil (TA), Thai (TH),

---

[4]For both investigated tasks, we find that `code-davinci-002` generally produces competitive or better results than `text-davinci-002` on a small set of samples. In consideration of budget, we choose to use `code-davinci-002` because it supports free access at the time of our experiment.

[5]https://github.com/cambridgeltl/xcopa

| Model | AVG | BG | DA | DE | ET | FA | FR | HR | IT | JA | KO | NL | ZH |
|---|---|---|---|---|---|---|---|---|---|---|---|---|---|
| HUMAN | – | 87.0 | – | 74.0 | – | 97.0 | – | – | 78.0 | 75.0 | 76.0 | – | 85.0 |
| XLM-R Large | **68.9** | **66.5** | **71.1** | 65.8 | **68.7** | **75.3** | 62.5 | **72.3** | **64.9** | 63.8 | 69.6 | **72.8** | **73.2** |
| Codex (code-davinci-002) | | | | | | | | | | | | | |
| DIRECT | 60.8 | 59.2 | 59.6 | 68.2 | 59.0 | 58.0 | 58.6 | 65.7 | 55.4 | 56.0 | 62.0 | 64.8 | 63.0 |
| EN-COT | 61.4 | 60.2 | 66.6 | 70.6 | 60.3 | 63.6 | **64.6** | 61.0 | 54.2 | 52.2 | 56.6 | 62.8 | 64.0 |
| PaLM-540B | | | | | | | | | | | | | |
| DIRECT | 66.7 | 62.6 | 67.4 | **72.6** | 62.3 | 75.0 | **64.6** | 65.0 | 59.4 | **64.0** | **70.2** | 72.0 | 64.8 |
| EN-COT | 63.2 | 63.4 | 64.6 | 68.6 | 61.5 | 67.2 | **64.6** | 55.9 | 57.4 | 55.6 | 66.4 | 69.4 | 64.0 |

Table 6: Accuracy on the XL-WiC languages with MULTILINGUAL-EXEMPLARS. XLM-R Large denotes the previous state-of-the-art results trained with 5.4K English examples (Raganato et al., 2020). The best model result in each column is in **boldface**.

Turkish (TR), Vietnamese (VO), and Mandarin Chinese (ZH). The task objective is to determine the causal relationship between the premise and two options based on a question (which is either "What was the *cause*?" or "What happened as a *result*?"). A successful model is, therefore, expected to not only perform commonsense reasoning but also generalize its reasoning capabilities to new languages. For each target language, XCOPA contains 100 annotated examples in the validation set and 500 examples in the test set. In our experiments, we focus on the examples in the test sets and use the ones in the validation set as few-shot exemplars whenever needed.

We test the Codex and PaLM models under both DIRECT and EN-COT. In both settings, we include the same set of examples, randomly selected from the validation sets of TR, ZH, TA, and QU, but for EN-COT, we additionally write brief rationales (in English) before the final answers ourselves.

**Results.** Table 5 presents our main results, along with per-language breakdowns for each XCOPA language. The previous state-of-the-art performance was around 76%, obtained by RoBERTa Large in the translate-test setting where the English RoBERTa Large model was first trained on the English COPA (Gordon et al., 2012) and English SIQa (Sap et al., 2019) datasets and then applied to the XCOPA test data, which was translated to English (Ponti et al., 2020). With only four multilingual chain-of-thought examples (EN-COT), PaLM-540B outperforms RoBERTa Large by a significant margin (14%), thereby setting a new high bar on XCOPA. While Codex performs better than RoBERTa Large, it still falls 9% behind PaLM-540B. We also highlight that PaLM-540B performs noticeably better than all the other models on under-represented languages such as ET, HT, and SW; this result suggests that PaLM-540B might have some internal knowledge about these languages.

## 5.2 XL-WiC

XL-WiC is a multilingual word in-context semantic judgment benchmark covering thirteen languages:[6] Bulgarian (BG), Danish (DA), German (DE), Estonian (ET), Persian (FA), French (FR), Croatian (HR), Italian (IT), Japanese (JA), Korean (KO), Dutch (NL) and Chinese (ZH). Given two sentences in the same language and a word of interest which appears in both sentences, the model is asked whether the word is of the same sense in the sentences. In order to arrive at the correct answer, a model needs to be aware of the concept of word sense, and to infer the sense of a word based on its context. Despite its simplicity, this task is extremely challenging; PaLM-540B only achieves a score of 64.6 on WiC (Pilehvar & Camacho-Collados, 2019), the English version of the task.

**Results.** We evaluate the cross-lingual word-in-context sense judgment performance of models (Table 6). With the supervision from only four examples, PaLM-540B achieves competitive or better results that the state-of-the-art model (XLM-R Large) on 6 (German, Persian, French, Japanese, Korean and Dutch) of the 12 investigated languages. However, we do not observe an improvement over direct answer prediction when using chain-of-thought prompting on this task.[7]

---

[6] https://pilehvar.github.io/xlwic/

[7] One potential reason is that our prompts are not necessarily optimal (Wang et al., 2022) and may benefit from a broader investigation of other prompt formats. On the other hand, rationales for this task are fairly

## 6 RELATED WORK

**Prompting.** Existing work (Radford et al., 2019; Brown et al., 2020; Schick & Schütze, 2021, *inter alia*) has shown that prompting pre-trained large language models can lead to strong performance on various tasks such as text classification (Shin et al., 2020; Gao et al., 2021), question answering (Khashabi et al., 2020), and program synthesis (Austin et al., 2021; Nye et al., 2021; Shi et al., 2022a): taking a few examples of the task in a certain pattern as the prompting input, models are often able to generate accurate output following the pattern. Wei et al. (2022b) have shown that chain-of-thought prompting significantly improves the reasoning performance of language models, by adding explicit reasoning steps before the final answer. Ahn et al. (2022) apply chain-of-thought prompting in robotics scenarios, including a multilingual setting. In this work, we systematically analyze multilingual few-shot chain-of-thought prompting on complicated reasoning benchmarks.

**Multilingual pre-trained language models.** Through masked language modeling (Devlin et al., 2019; Conneau et al., 2020), auto-regressive language modeling (Brown et al., 2020; Ouyang et al., 2022) or encoder-decoder training (Liu et al., 2020; Chen et al., 2021; Xue et al., 2021), pre-trained Transformer-based large language models have shown impressive performance on multiple NLP tasks across languages. Previous work (Zhao & Schütze, 2021; Winata et al., 2021; Lin et al., 2021b) investigated prompting in the multilingual setting and found that using English prompts with non-English examples led to strong few-shot performance. Evaluation of multilingual models has mostly focused on general information extraction tasks such as question answering (Clark et al., 2020; Hu et al., 2020; Kassner et al., 2021; Ruder & Sil, 2021) as well as specific types of reasoning such as commonsense reasoning (Ponti et al., 2020; Lin et al., 2021a) and temporal reasoning (Ruder et al., 2021). To the best of our knowledge, this is the first study to evaluate the multilingual multi-step reasoning abilities of large language models.

**Cross-lingual transfer and generalization.** Previous work has demonstrated that pre-trained multilingual models significantly help cross-lingual transfer on a wide range of NLP tasks such as cross-lingual named entity recognition (Pires et al., 2019; Mulcaire et al., 2019), zero-shot cross-lingual dependency parsing (Schuster et al., 2019; Shi et al., 2022b), and bilingual lexicon induction (Shi et al., 2021). In this work, we demonstrate strong cross-lingual generalization of PaLM (§4.2, §5) and Codex (§5), on three tasks that require complicated reasoning.

**Multilingual benchmarks.** To test the multilingual NLP performance of existing models, there has been work introducing benchmarks on various multilingual tasks, including cross-lingual question answering (Liu et al., 2019; Clark et al., 2020), natural language inference (Conneau et al., 2018) and bilingual lexicon induction (Lample et al., 2018), as well as collections across tasks (Hu et al., 2020; Ruder et al., 2021). The tasks in these multilingual benchmarks, to the best of our knowledge, require relatively simple reasoning processes. In this paper, we present MGSM, a multilingual arithmetic reasoning benchmark, which can be used to test multilingual multi-step reasoning abilities of models.

## 7 CONCLUSION

In this paper, we introduce MGSM, the first multilingual benchmark to evaluate arithmetic reasoning abilities of language models. MGSM is an extension of the GSM8K dataset (Cobbe et al., 2021) and contains 250 examples written in *ten* typologically diverse languages. We also present a comprehensive analysis of the multilingual reasoning abilities of large language models such as GPT-3 and PaLM on multiple multilingual benchmarks, including our own MGSM dataset. We find that large-scale language models appear to perform complex multi-step reasoning across multiple languages, including those underrepresented languages which are covered by less than $0.01\%$ of training corpora. Finally, we demonstrate that multilingual chain-of-thought prompting is an empirically effective approach to multilingual commonsense reasoning, outperforming the previous best model on the challenging XCOPA dataset by 13% on average.

---

straight-forward and example-specific. It is thus unclear whether the WiC task requires true reasoning that benefits from the depiction of intermediate reasoning steps. We leave further investigation for future work.

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

| | en | de | fr | es | ru | zh | ja | th | te | bn | sw |
|---|---|---|---|---|---|---|---|---|---|---|---|
| # Exemplars | 6 | 6 | 6 | 6 | 1 | 5 | 4 | 1 | 1 | 1 | 6 |

Table 7: Number of few-shot exemplars for GPT-3 experiments in Table 3.

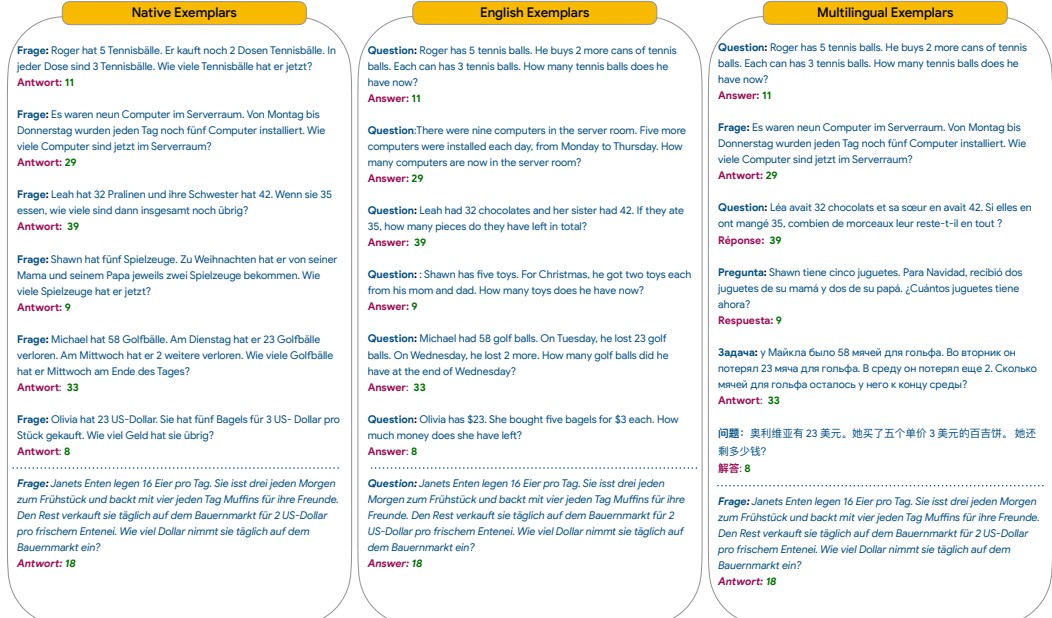

Figure 6: Prompt template in the direct answer prediction setting (DIRECT), solving a problem in German. Above dotted lines: few-shot exemplars; below dotted lines: the question of interest and the expected answer. The dotted lines are not included in our experiments.

## A DETAILS OF MGSM EXPERIMENTS

In this section, we present details of our experiments on MGSM, including the number of exemplars used for GPT-3 (§A.1) and the detailed prompts in each setting summarized in Table 2 (§A.2).

### A.1 NUMBER OF EXEMPLARS FOR EACH LANGUAGE

Given the unbalanced representation of languages in the training corpora, the byte-pair encoding (BPE; Gage, 1994) algorithm tokenizes sentences in underrepresented languages, especially those in a different alphabet from English, into more tokens. Given that the GPT-3 API supports a maximum number of 2048 tokens as its input, it does not support 6-shot prompting in some languages, including Russian, Chinese, Japanese, Thai, Telugu and Bengali; therefore, we use the maximum possible number of exemplars (Table 7) instead for GPT-3, while using 6-shot for all languages in PaLM experiments.

### A.2 MGSM PROMPTS IN EACH SETTING

We present the prompts used in our MGSM experiments in Figures 6 to 8, where the TRANSLATE-EN experiments can be viewed as a English one with EN-CoT and ENGLISH-EXEMPLARS.

## B DETAILED MGSM PERFORMANCE

We report the detailed numbers in our analysis (Figures 4 and 5) in Table 8.

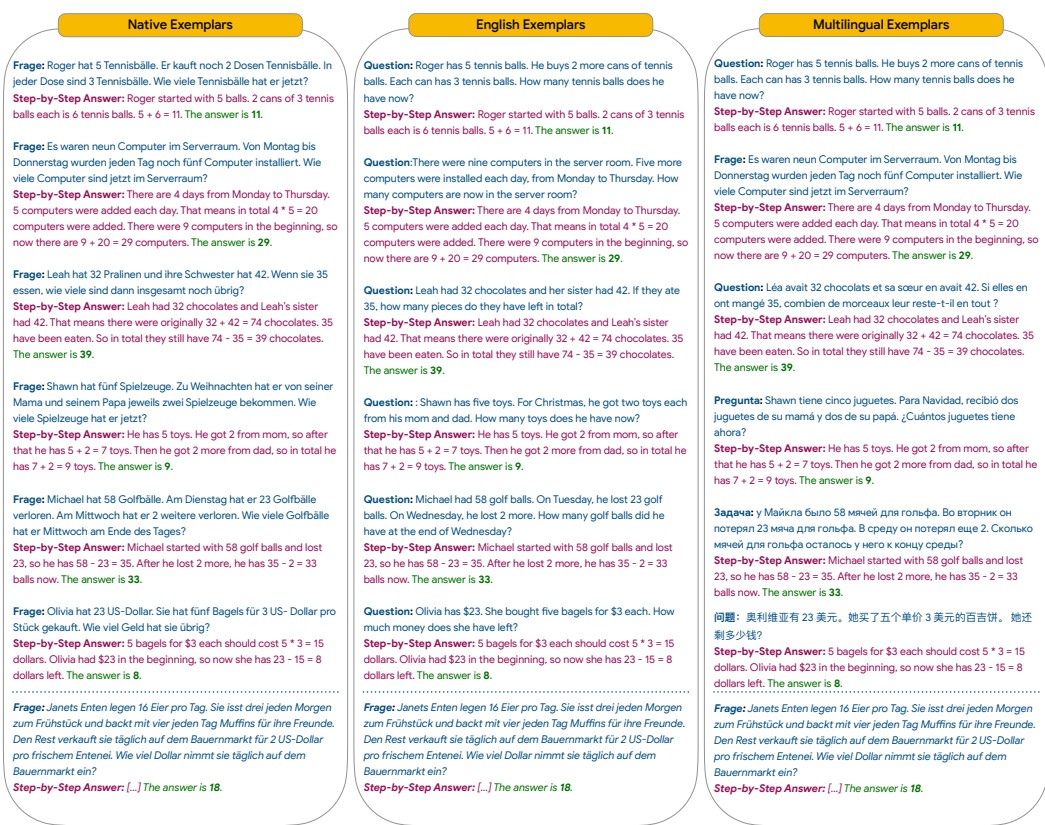

Figure 7: Prompt template in the English CoT setting (EN-CoT), solving a problem in German. Above dotted lines: few-shot exemplars; below dotted lines: the question of interest and the expected answer. The dotted lines are not included in our experiments.

**Native Exemplars**

**Frage:** Roger hat 5 Tennisbälle. Er kauft noch 2 Dosen Tennisbälle. In jeder Dose sind 3 Tennisbälle. Wie viele Tennisbälle hat er jetzt?
**Schritt-für-Schritt-Antwort:** Roger begann mit 5 Bällen. 2 Dosen von jeweils 3 Tennisbällen macht 6 Tennisbälle. 5 + 6 = 11. Die Antwort ist **11**.

**Frage:** Es waren neun Computer im Serverraum. Von Montag bis Donnerstag wurden jeden Tag noch fünf Computer installiert. Wie viele Computer sind jetzt im Serverraum?
**Schritt-für-Schritt-Antwort:** Von Montag bis Donnerstag sind es 4 Tage. Jeden Tag kamen 5 neue Computer hinzu. Das macht insgesamt 4 x 5 = 20 Computer, die hinzugefügt wurden. Am Anfang waren es 9 Computer, also sind es jetzt 9 + 20 =29 Computer. Die Antwort lautet **29**.

**Frage:** Leah hat 32 Pralinen und ihre Schwester hat 42. Wenn sie 35 essen, wie viele sind dann insgesamt noch übrig?
**Schritt-für-Schritt-Antwort:** Leah hat 32 Pralinen und Leahs Schwester 42. Das bedeutet, dass es ursprünglich 32 + 42 =74 Pralinen waren. 35 wurden gegessen. Also haben sie insgesamt noch 74 - 35 = 39 Pralinen übrig. Die Antwort lautet **39**.

**Frage:** Shawn hat fünf Spielzeuge. Zu Weihnachten hat er von seiner Mama und seinem Papa jeweils zwei Spielzeuge bekommen. Wie viele Spielzeuge hat er jetzt?
**Schritt-für-Schritt-Antwort:** Er hat 5 Spielzeuge. Er hat 2 von seiner Mama bekommen, sodass er nun 5 + 2 = 7 Spielzeuge hat. Dann hat er noch 2 von seinem Papa bekommen, also hat er insgesamt 7 + 2 = 9 Spielzeuge. Die Antwort lautet **9**.

**Frage:** Michael hat 58 Golfbälle. Am Dienstag hat er 23 Golfbälle verloren. Am Mittwoch hat er 2 weitere verloren. Wie viele Golfbälle hat er Mittwoch am Ende des Tages?
**Schritt-für-Schritt-Antwort:** Michael hatte anfangs 58 Golfbälle und hat 23 verloren, sodass er 58 - 23 = 35 hat. Nachdem er 2 weitere verloren hat, hat er jetzt 35 - 2 = 33 Bälle. Die Antwort lautet **33**.

**Frage:** Olivia hat 23 US-Dollar. Sie hat fünf Bagels für 3 US- Dollar pro Stück gekauft. Wie viel Geld hat sie übrig?
**Schritt-für-Schritt-Antwort:**5 Bagels für 3 US-Dollar pro Stück kosten 5 x 3 = 15 Dollar. Olivia hat anfangs 23 US-Dollar, also hat sie jetzt 23 - 15 = 8 Dollar übrig. Die Antwort lautet **8**.

......................................................................

*Frage: Janets Enten legen 16 Eier pro Tag. Sie isst drei jeden Morgen zum Frühstück und backt mit vier jeden Tag Muffins für ihre Freunde. Den Rest verkauft sie täglich auf dem Bauernmarkt für 2 US-Dollar pro frischem Entenei. Wie viel Dollar nimmt sie täglich auf dem Bauernmarkt ein?*
*Schritt-für-Schritt-Antwort: [...] Die Antwort lautet **18**.*

**Multilingual Exemplars**

**Question:** Roger has 5 tennis balls. He buys 2 more cans of tennis balls. Each can has 3 tennis balls. How many tennis balls does he have now?
**Step-by-Step Answer**: Roger started with 5 balls. 2 cans of 3 tennis balls each is 6 tennis balls. 5 + 6 = 11. The answer is **11**.

**Frage:** Es waren neun Computer im Serverraum. Von Montag bis Donnerstag wurden jeden Tag noch fünf Computer installiert. Wie viele Computer sind jetzt im Serverraum?
**Schritt-für-Schritt-Antwort:** Von Montag bis Donnerstag sind es 4 Tage. Jeden Tag kamen 5 neue Computer hinzu. Das macht insgesamt 4 x 5 = 20 Computer, die hinzugefügt wurden. Am Anfang waren es 9 Computer, also sind es jetzt 9 + 20 =29 Computer. Die Antwort lautet **29**.

**Question:** Léa avait 32 chocolats et sa sœur en avait 42. Si elles en ont mangé 35, combien de morceaux leur reste-t-il en tout ?
**Réponse étape par étape:** Léa avait 32 chocolats et sa sœur en avait 42. Cela signifie qu'il y avait à l'origine 32 + 42 = 74 chocolats. 35 chocolats ont été mangés. Il leur en reste donc au total 74 - 35 = 39 chocolats. La réponse est **39**.

**Pregunta:** Shawn tiene cinco juguetes. Para Navidad, recibió dos juguetes de su mamá y dos de su papá. ¿Cuántos juguetes tiene ahora?
**Respuesta paso a paso:** Tiene 5 juguetes. Recibió 2 de la mamá, por lo que después de eso tiene 5 + 2 = 7 juguetes. Luego, recibió 2 más del papá, así que en total tiene 7 + 2 = 9 juguetes. La respuesta es **9**.

**Задача:** у Майкла было 58 мячей для гольфа. Во вторник он потерял 23 мяча для гольфа. В среду он потерял еще 2. Сколько мячей для гольфа осталось у него к концу среды?
**Пошаговое решение**: вначале у Майкла было 58 мячей для гольфа, 23 он потерял, и у него осталось 58 - 23 = 35. После этого он потерял еще 2, и теперь у него осталось 35 - 2 = 33 мяча. Ответ — **33**.

**问题:** 奥利维亚有 23 美元。她买了五个单价 3 美元的百吉饼。她还剩多少钱
**逐步解答**: 5 个单价 3 美元的百吉饼应该花费 5 * 3 = 15 美元。奥利维亚一开始有 23 美元，所以现在她还剩 23 - 15 = 8 美元。答案是 **8**。

......................................................................

*Frage: Janets Enten legen 16 Eier pro Tag. Sie isst drei jeden Morgen zum Frühstück und backt mit vier jeden Tag Muffins für ihre Freunde. Den Rest verkauft sie täglich auf dem Bauernmarkt für 2 US-Dollar pro frischem Entenei. Wie viel Dollar nimmt sie täglich auf dem Bauernmarkt ein?*
*Schritt-für-Schritt-Antwort: [...] Die Antwort lautet **18**.*

Figure 8: Prompt template with CoT in the question language (NATIVE-COT), solving a problem in German. Above dotted lines: few-shot exemplars; below dotted lines: the question of interest and the expected answer. The dotted lines are not included in our experiments.

|  | AVG | HRL | LRL | EN | DE | FR | ES | RU | ZH | JA | TH | TE | BN | SW |
|---|---|---|---|---|---|---|---|---|---|---|---|---|---|---|
| Lang. freq. (%) | - | - | - | 78.0 | 3.5 | 3.3 | 2.1 | 0.53 | 0.40 | 0.38 | 0.04 | 0.02 | 0.006 | 0.005 |
| PaLM |  |  |  |  |  |  |  |  |  |  |  |  |  |  |
| • Exemplar token length (avg.) |  |  |  | 95 | 108 | 119 | 105 | 113 | 118 | 118 | 193 | 199 | 173 | 130 |
| • NATIVE-COT |  |  |  |  |  |  |  |  |  |  |  |  |  |  |
| - 8B 6-shot | 4.0 | 4.1 | 3.1 | 6.4 | 6.8 | 4.4 | 2.4 | 2.8 | 4.0 | 4.4 | 3.2 | 3.6 | 3.2 | 2.4 |
| - 62B 6-shot | 20.0 | 22.7 | 13.2 | 30.4 | 24.0 | 24.0 | 26.0 | 22.8 | 24.8 | 14.8 | 18.0 | 11.6 | 13.6 | 9.6 |
| - 540B 1-shot | 38.9 | 39.7 | 34.8 | 50.8 | 42.8 | 44.8 | 44.8 | 41.2 | 34.8 | 29.6 | 40.0 | 38.0 | 34.0 | 27.2 |
| - 540B 2-shot | 43.7 | 44.0 | 39.8 | 57.2 | 47.2 | 43.2 | 50.4 | 44.4 | 44.4 | 34.4 | 47.2 | 38.0 | 40.8 | 33.2 |
| - 540B 4-shot | 45.1 | 45.5 | 41.0 | 58.8 | 44.8 | 49.6 | 47.6 | 46.4 | 46.4 | 38.4 | 46.4 | 41.2 | 44.4 | 32.0 |
| - 540B 6-shot | 48.1 | 47.9 | 44.9 | 62.4 | 49.2 | 46.4 | 56.8 | 48.4 | 46.8 | 40.0 | 52.8 | 45.6 | 46.0 | 35.2 |

Table 8: Detailed performances corresponding to Figures 4 and 5.

## C    THE CHAIN-OF-THOUGHT PROMPTS USED IN THE PAPER

In this section, we present the details of the chain-of-thought prompts used in our paper for the XCOPA (Figure 9) and the XL-WiC (Figures 10 and 11) tasks.

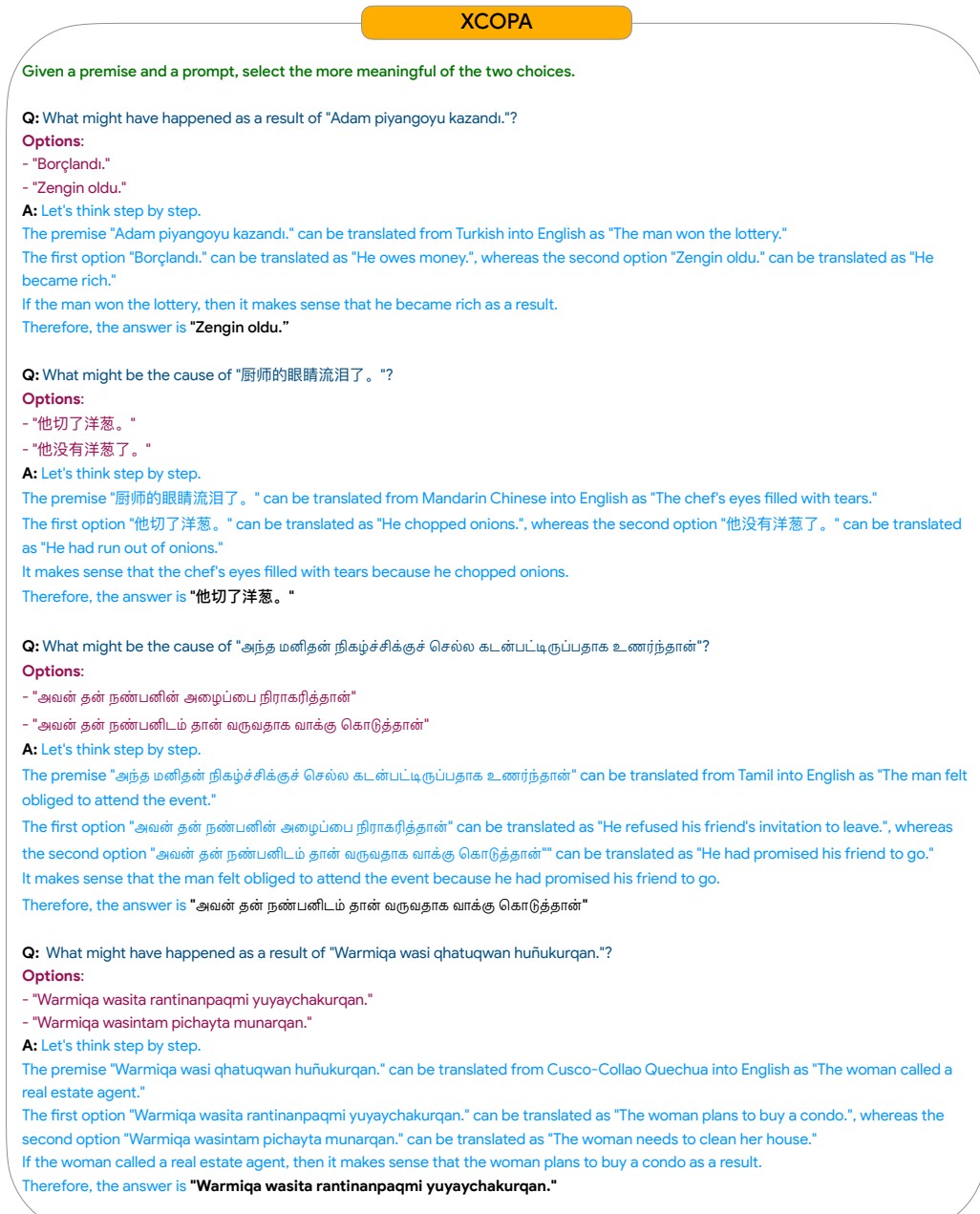

Figure 9: The chain-of-thought prompt used in the XCOPA experiments. The four examples are randomly selected from the validation sets of Turkish (TR), Mandarin Chinese (ZH), Tamil (TA), and Cusco-Collao Quechua (QU). The rationales are written by the authors, and the task description is taken directly from (Ponti et al., 2020). Under the direct prompting setup, the answers (**bolded**) are given directly and rationales are entirely omitted.

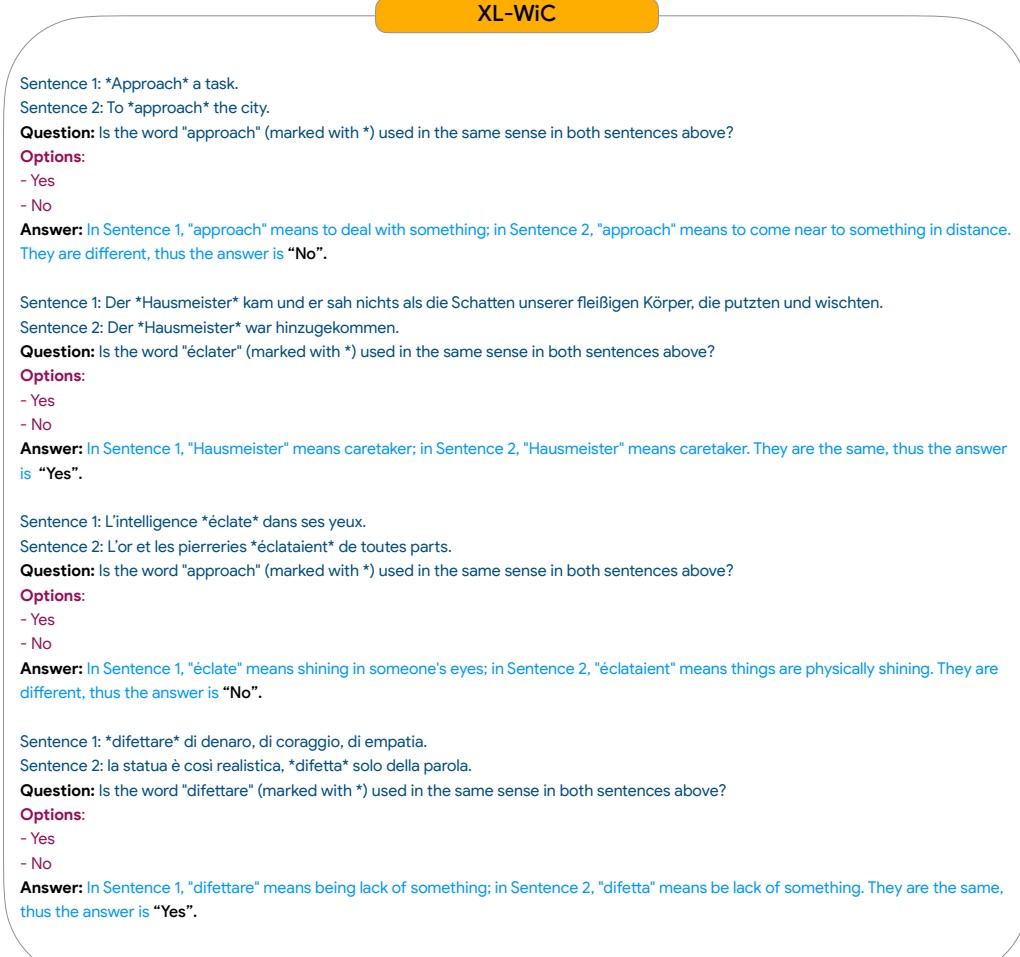

Figure 10: The multilingual chain-of-thought prompt used in the XL-WiC experiments.

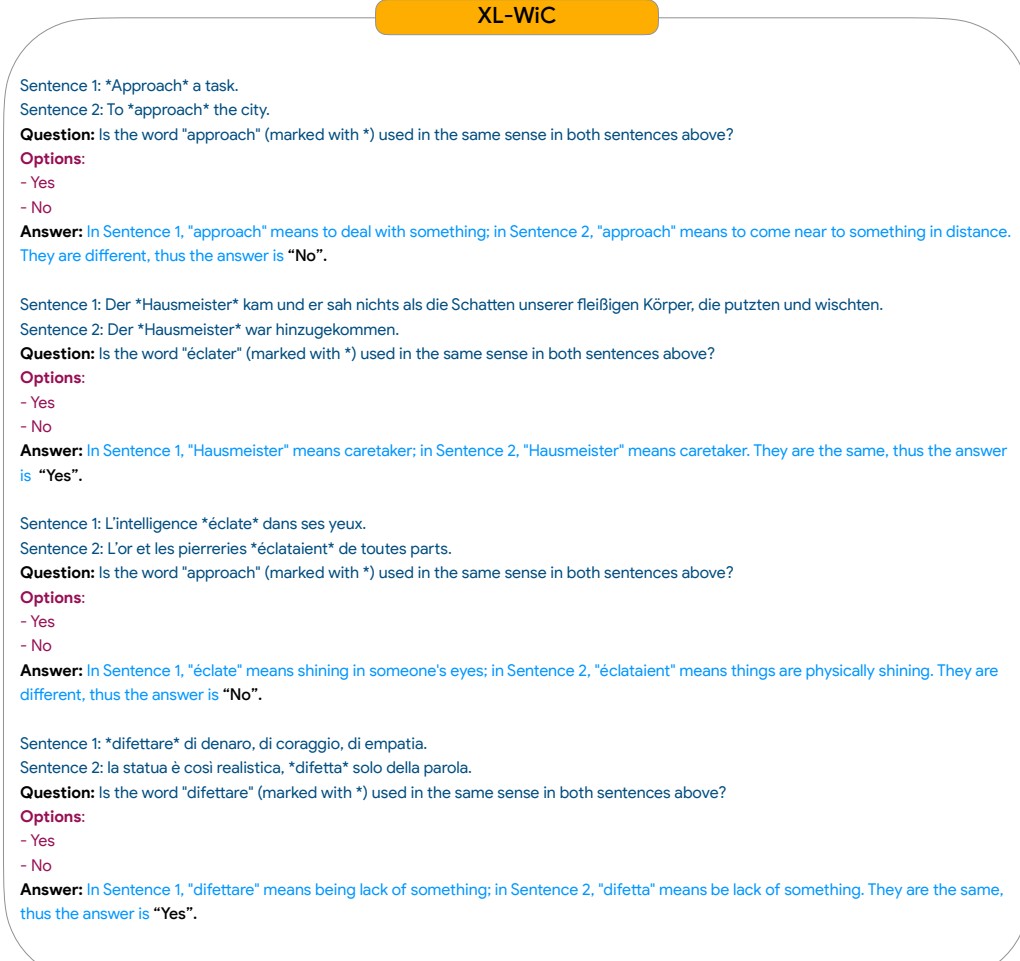

Figure 11: The English-language chain-of-thought prompt used in the XL-WiC experiments.

