# OpenReview forum: "Language models are multilingual chain-of-thought reasoners"
_ICLR.cc/2023/Conference — ICLR 2023 poster_

### Official Review · Reviewer_DTUx · 2022-10-24

**Confidence:** 3
**Correctness:** 4
**Technical Novelty And Significance:** 2
**Empirical Novelty And Significance:** 4
**Recommendation:** 6

**Clarity, Quality, Novelty And Reproducibility:**

This paper is well-written and easy to follow. The dataset is novel while the proposed multilingual prompting strategy is incremental to villa chain-of-thought. Considering the heavy resource requirement of experiment, it might be difficult for reproduce given limited resources.

**Strength And Weaknesses:**

Strength:
1.	The MGSM is the first multilingual dataset that can be used to evaluate the chain of thought in multilingual setting.
2.	The paper is well-writing and easy to read.
3.	The experiments are comprehensive and well organized.
Weakness:
1.	The contribution of multilingual chain-of-thought is incremental compared to the villa chain-of-though.
2.	It is interesting to see the large gap between Translate-EN, EN-CoT and Native-CoT in MGSM. While the gaps in other benchmarks are not too much. Is it because MGSM benchmark is originated from translation?


**Summary Of The Paper:**

This paper introduces a Multilingual Grade School Math (MGSM) which can be used to evaluate the reasoning abilities of large language models by predicting the chain of thought. The authors compare the results of different prompting strategies on two large-scale language models, and further conduct ablation studies to analyze the effect of language frequency, model scale, exemplar amount and exemplar type.

**Summary Of The Review:**

The contribution of the new benchmark is clear and the experiments are sufficient, while the prompting strategy of multilingual chain-of-thought is incremental.

---

> ### Author Response · Authors · 2022-11-17
> **Response to Reviewer DTUx**
>
> We thank the reviewer for the positive feedback and valuable questions!  We are delighted to learn that the reviewer has found our experiments to be comprehensive and well-organized. In what follows, we would like to provide further clarifications about our work based on their questions and comments, while addressing the novelty concerns:
>
> ### On Contributions of This Work
> We would like to highlight that our proposed contribution is not the mere usage of multilingual chain-of-thought per se. We agree with the reviewer that multilingual CoT is a simple but effective extension of vanilla CoT. However, we believe that our primary contributions are (i) the introduction of MGSM as the first multilingual corpus for arithmetic reasoning (covering ten typologically diverse languages), (ii) demonstration of the multilingual capabilities of large language models such as GPT-3 and LLM, and (iii) extensive investigation of the multilingual arithmetic and commonsense reasoning capabilities of large language models through experimentations on XCOPA and WiC. We show, among other findings, that multilingual CoT, with a few illustrative few-shot examples, enables these models to perform on par, and sometimes even outperform, fully supervised and fully specialized multilingual models. Therefore, the contributions are more on the dataset and phenomenon side than on the methodology (usage of multilingual CoT). We will endeavor to make this point clearer in the next iteration of the manuscript.
>
> ### Gap Between Translate-EN-CoT, EN-CoT, and Native-CoT in MGSM
> It is indeed interesting to observe gaps in the performances between the Translate-EN-CoT, EN-CoT, and Native-CoT methods on MGSM. We speculate that these models, especially the GPT-3 model, are more apt at performing intermediate reasoning steps in English than in other languages; therefore, EN-CoT and Translate-EN-CoT seem to perform often better than Native-CoT. That said, we would like to note that these gaps are most pronounced in the case of GPT-3 (text-davinci-002). On average, for instance, Translate-EN-CoT is almost 19% better than Native-CoT and 14% better than EN-CoT for GPT-3. On the other hand, we note that these gaps are much smaller for LLM-540B. For instance, on average, Translate-EN-CoT is almost 7% better than Native-CoT and 4% better than EN-CoT—and the accuracy numbers are significantly higher for LLM-540B than for GPT-3. To answer the reviewer’s second question, we believe that these gaps in performances in MGSM do not necessarily stem from the fact that MGSM was translated from English. (XCOPA too was originally translated from English, for instance.) We hope that our responses answer the reviewer's questions.
>
> ### On Reproducibility
> We appreciate the reproducibility concerns of the reviewer; however, we are delighted to share that we already provided the full set of the direct and chain-of-thought prompts used in our experiments. And we will release our supplementary materials upon the publication of our work so that researchers can make meaningful comparisons with our results.

---

### Official Review · Reviewer_q52S · 2022-10-24

**Confidence:** 4
**Correctness:** 4
**Technical Novelty And Significance:** 4
**Empirical Novelty And Significance:** 4
**Recommendation:** 8

**Clarity, Quality, Novelty And Reproducibility:**

Clarity: very clear, this is a straightforward analysis but delivered in a very clear way

Quality: very good, the experiments are thoroughly done

Novelty: high. I wonder why nobody else did this until now..... </sarcasm>

Reproducibility: as soon as the 540B model will be release or an API provided this can be reproduced very easily

**Strength And Weaknesses:**

STRENGTH
1. Translating a math-problem dataset (GSM8K) from English to 10 other diverse languages, using professional translators; and releasing the full dataset
2. Benchmarking two LLMs with CoT on that and two other multi-lingual datsets; showing that they provide competitive and sometimes state-of-the-art results compared to a fully supervised setting
3. Showing that machine-translating the CoT into English is at least as good and often better than prompting with CoT in the target language directly

WEAKNESS
1. A point which is never addressed is why translated CoT seems to perform always better than human translated CoT. Is this because some of the training data of the LLM might be translatese? Or is the difference too small to be significant? If so, it would be interesting to see the results with a different set of prompts or with a different sampling
2. Not really a weakness, but this result will not necessarily encourage more research into low-ressource languages. While the large LLM in this paper is anonymized, I picked a random other language model from all those that also have 540B (PALM in this case). PALM contained 24B French tokens, and this paper reports 46.2% when natively prompted and 55.2% when translated. Swahili only had 40M tokens for PALM, but has a comparable performance in this paper when translated (51.2%, compared to 35.2% for native Swahili prompt). Why developping a Swahili (or French?) LLM when I can just use the largest EN one and machine-translate? At the same time, 40M tokens will have much less variety than 578B tokens used for English and some idiosyncrasy of Swahili might not be captured by translation only.
Another direction which will be encouraged by this paper is scaling: Swahili had an almost 4x jump between using a 62B model (there are a few of those available these days) and a 540B (not so many available of that size).

Note that Weakness (2) is not really an issue with the paper... just with the consequences that the reported facts might produce.



**Summary Of The Paper:**

This is a very straightforward paper, with impressive and surprising - although to me sad (more on this later) - results. They will certainly motivate future work, and encourage some of the current directions.

The authors analyze the multi-lingual capabilities of language models who are trained primarily on English data, with a little of other languages as well. They combine this with recent work of chain-of-thought (CoT) prompting, showing that those the combination of large LLM + CoT provide a very decent results on challenging datasets, in a multi-lingual setting.

**Summary Of The Review:**

Straightforward idea, very well executed. High impact

---

> ### Author Response · Authors · 2022-11-17
> **Response to Reviewer q52S**
>
> We thank the reviewer for the valuable feedback and valuable suggestions! We are delighted to see that the reviewer found our paper to be clear and of good quality. In what follows, we would like to provide further clarifications about our dataset based on their questions and comments:
>
> ### On Different CoT Prompting Strategies
>
> > A point which is never addressed is why translated CoT seems to perform always better than human translated CoT. Is this because some of the training data of the LLM might be translatese? Or is the difference too small to be significant? If so, it would be interesting to see the results with a different set of prompts or with a different sampling
>
> We apologize for the source of this confusion, but we would like to mention that we do not have a CoT strategy in which we are using human-translated explanations. In our main experiments, we focused on three CoT strategies, namely Native-CoT, EN-CoT, and Translate-EN. In Native-CoT, we solve the problem in the native language of the question (by predicting the intermediate steps in the original source language of the problem). In EN-CoT, we expect the model to generate the chain-of-thought steps in English (please see the second plot in Figure 7 in the Appendix for an illustration). Finally, in Translate-EN-CoT, given a problem at inference time, we use the Google Translate API to translate the problem to English and then input the translated problem in English to the model and expect it to generate a step-by-step chain-of-thought solution in English. We wanted to make this clarification, because we didn’t have a human-translated CoT baseline. However, if the reviewer is curious about why English-based CoT methods might be performing relatively better than non-English-based CoT methods such as Native-CoT, we speculate that these large language models are more apt at generating CoT-explanations in English than in other languages, as English is the dominant language in the pre-training corpora used in the pretraining of these large-scale models.
>
> ### Encouraging More Research into Low-Resource Languages
> > Not really a weakness, but this result will not necessarily encourage more research into low-ressource languages. While the large LLM in this paper is anonymized, I picked a random other language model from all those that also have 540B (PALM in this case). PALM contained 24B French tokens, and this paper reports 46.2% when natively prompted and 55.2% when translated. Swahili only had 40M tokens for PALM, but has a comparable performance in this paper when translated (51.2%, compared to 35.2% for native Swahili prompt). Why developping a Swahili (or French?) LLM when I can just use the largest EN one and machine-translate? At the same time, 40M tokens will have much less variety than 578B tokens used for English and some idiosyncrasy of Swahili might not be captured by translation only. Another direction which will be encouraged by this paper is scaling: Swahili had an almost 4x jump between using a 62B model (there are a few of those available these days) and a 540B (not so many available of that size).
>
> We truly appreciate the insightful remarks of the reviewer. Our work aims to encourage more research in underrepresented languages in multiple ways. First, as most of the NLP benchmarks are formulated in English-language, we hope that MGSM, like the multilingual common sense reasoning dataset XCOPA by Ponti et al. (2020), will encourage researchers to consider performing experiments and analysis on other languages. Second, we didn’t know, prior to the completion of this work, much about the multilingual reasoning and cross-transfer learning capabilities of these large-scale language models. We hope that our positive findings will encourage researchers to explore more about the ways in which they can use these models to conduct multilingual NLP research.
> Third, as the reviewer also hinted at, our work highlights the need for training a large-scale language model which makes use of corpora based on a variety of world languages.
>
> ### Reproducibility
> We are delighted to share that we already included the full set of the direct and chain-of-thought prompts used in our experiments, in the supplementary materials. Others can now reproduce our results on OpenAI models (such as text-davinci-002 and code-davinci-002), and we will release our supplementary materials upon the publication of our work so that researchers can make meaningful comparisons with our results.

---

### Official Review · Reviewer_aQqu · 2022-11-01

**Confidence:** 2
**Correctness:** 4
**Technical Novelty And Significance:** 2
**Empirical Novelty And Significance:** 3
**Recommendation:** 6

**Clarity, Quality, Novelty And Reproducibility:**

Clarity:
The claim is clear and backed by experiments.

Quality:
The paper is well written. The quality is ok but not extremely solid. Experimental investigations do not cover necessary scope.

Novelty:
The novelty is limited. The technique is merely another verification of existing published paper CoT with only minimum adaptation, where a naive translation of examples provides the best performance.

Reproducibility:
The dataset is released and the code is available. Although I assume most people do not have access to GPT3 or the refered 540B LLM, which are the only keys to obtain such SoTA results.

**Strength And Weaknesses:**

Strength:

1. The paper is well presented and easy to understand.

2. a multilingual version of the popular benchmark GSM8K is proposed and released, which might be of interest to the community.

3. new state-of-the-art on XCOPA and XLWiC is set.

Weaknesses:

1. The paper does not provide much novelty, insights and inspiration, the technique design simply follows existing works of CoT.

2. The empirical investigation is not sufficient for a solid benchmarking of the new datasets. Specifically, I believe several recently-proposed methods like ZeroShotCoT[1] and SelfConsistencyCoT[2] are also necessary and not difficult to implement.

3. The extension of the multilingual version of an existing benchmark seems providing only limited contribution for a ICLR paper to me. And the empirical verification of CoT on more datasets are also not sufficient for another new paper.

[1] Large Language Models are Zero-Shot Reasoners

[2] Self-Consistency Improves Chain of Thought Reasoning in Language Models

**Summary Of The Paper:**

This paper extends the prevalent GSM8K benchmark to a multilignual verison covering ten languages, and accordingly verify a recently proposed technique Chain-of-though upon it, resulting a best performance of 55 with the most competent LLM.

The authors also verify such capability on established benchmark XCOPA, achieving a new state-of-the-art.

The paper is well presented and very clear to understand.

**Summary Of The Review:**

The paper is well presented, an multilingual extension of GSM8K benchmark is proposed, CoT is verified accordingly as well as more datasets like XCOPA, which sets a new SoTA.

But the technique is almost identical to CoT, and the results can be well expected, leaving it only a very incremental contribution. Besides, such empirical verification is also not sufficient as important baselines are missing.

Therefore, I vote for borderline reject, but I'm not absolutely sure and are open for discussion or rebuttal.

---

> ### Author Response · Authors · 2022-11-17
> **Response to Reviewer aQqu**
>
> We thank the reviewer for the critical comments and valuable suggestions! We are delighted to see that the reviewer found our paper to be clear and well-supported by experiments. In what follows, we would like to provide further clarifications about our work based on their questions and comments:
>
> ### Novelty and Contributions
> We would like to point out once again that the contributions of the paper are manyfold:
> - We introduce MGSM, the first multilingual benchmark designed to assess arithmetic reasoning capabilities of language models.
> - We evaluate two large-scale language models, namely GPT-3 and LLM, on MGSM and show that these models, when equipped with chain-of-thought prompting, can perform on par, and sometimes even outperform, fully supervised and fully specialized multilingual models.
>
> We understand and appreciate the concern of the reviewer regarding the size of the dataset, but we would like to point out that the other similar cross-lingual/multilingual benchmarks, such as XCOPA [1], are more or less of the same size. We agree with the reviewer that it would be desirable to have more examples for each language, but due to financial constraints, we were able to translate only 250 examples from GSM8K to ten typologically different languages with the help of professional translators. In the future, once we receive another fund, we hope to extend the scope and size of our benchmark and update our benchmark.
>
> > The extension of the multilingual version of an existing benchmark seems providing only limited contribution for a ICLR paper to me. And the empirical verification of CoT on more datasets are also not sufficient for another new paper.
>
> First of all, as we highlight in our manuscript, there is a need for having a multilingual corpus for arithmetic reasoning, and our present work aims to fill in that gap. There are currently a few multilingual datasets for common-sense reasoning or multitask cross-lingual transfer; however, no work, at least to the best of our knowledge has introduced a typologically diverse multilingual corpus for arithmetic reasoning, covering eleven languages (some of which include under-represented languages such as Swahili and Telugu).
>
>
> ### Zero-Shot Chain-of-Thought Baseline
> We appreciate the suggestions by the reviewer. During the author response period, we ran the zero-shot CoT experiments. We present our results below:
>
> |                                     |  avg |  hrl |  lrl |  en  |  de  |  fr  |  es  |  ru  |  zh  |  ja  |  th  |  te  |  bn  |  sw  |
> |-------------------------------------|:----:|:----:|:----:|:----:|:----:|:----:|:----:|:----:|:----:|:----:|:----:|:----:|:----:|:----:|
> | text-davinci-002 (EN-CoT-0shot)     | 29.5 | 37.2 | 13.7 | 46.8 | 40.0 | 36.0 | 42.4 | 40.0 | 33.2 | 31.6 | 19.6 |  6.8 | 15.2 | 13.2 |
> | text-davinci-002 (Native-CoT-0shot) |  9.8 |  7.8 |  3.6 | 46.8 |  8.4 |  4.8 | 13.2 |  6.4 |  5.2 |  8.8 |  5.2 |  1.2 |  2.4 |  5.6 |
> | LLM-540B (EN-CoT-0shot)             | 30.8 | 38.3 | 15.2 | 48.0 | 38.4 | 36.0 | 42.4 | 42.0 | 35.6 | 35.2 | 20.0 | 10.4 | 14.0 | 16.4 |
> | LLM-540B (Native-CoT-0shot)         | 14.4 | 13.2 |  7.7 | 48.0 | 12.8 | 12.4 | 16.8 | 13.6 | 10.8 | 12.8 |  7.6 |  6.8 |  6.8 |  9.6 |
>
> We've also had this updated in our manuscript.
>
> We note that self-consistency is a simple but computationally costly baseline, and it is a technique that is orthogonal to CoT: all of our investigated methods can be combined with self-consistency. Due to computational limitations, we couldn’t include it as a baseline in our results, but we hope to investigate its use in the future.

---

> > ### Comment · Reviewer_aQqu · 2022-12-12
> > **Reply**
> >
> > I appreciate the authors' effort in including Zero-shot results. Although I still believe Self-Consistency is a necessary baseline missed by this paper, I agree with the authors' rebuttal that it is an orthogonal method to this paper, and should have minimum impact on the arrived conclusion.
> >
> > As all other reviewers are now reached consistency, and my evaluation is initially given with a very low confidence, I am now accordingly increasing the score higher but still keeping a low confidence.
> >
> > I encourage the authors to include the Self-Consistency once they got enough funds in the future.

---

### Official Review · Reviewer_3ofN · 2022-11-02

**Confidence:** 4
**Correctness:** 4
**Technical Novelty And Significance:** 2
**Empirical Novelty And Significance:** 3
**Recommendation:** 6

**Clarity, Quality, Novelty And Reproducibility:**

Clarity: Very clear, the paper is well-written.
Quality: Good.
Novelty: This work is a multilingual version extension of the few-shot chain-of-thought reasoner. They empirically evaluated the multilingual reasoning capability of two large scale language models.

**Strength And Weaknesses:**

Strength:
1.  The paper shows interesting experimental results on the multilingual arithmetic reasoning task MGSM. Large-scale language models can perform well across languages, and even on underrepresented languages like Swahili.
2. The paper provides extensive and convincing experiment results on other two multilingual tasks (common sense reasoning and word in-context semantic judgment).

Weaknesses:
1. The constructed test sample size for the benchmark is a bit small (250 samples).
2. Some experiments and additional clarification can be considered and added (see Questions).

Questions:
1. In Section 4, it would be better if the authors add the experimental results of adopting Zero-Shot CoT (Kojima, Takeshi, et al. 2022) [1], compared with the Few-Shot CoT model.
2. One major question I have in mind is, is the current sample size of MGSM sufficient as a benchmark corpus? It would be better if the test sample size can be larger than the current 250, as the original GSM8K contains 1,000 test samples.
3. It would be better if authors can add some insight or feature-based analysis on the emergent ability of cross-lingual reasoning of the tested language models GPT-3 and LLM.
4. It is a bit surprising that the language models (GPT-3 and LLM) that are not specially pre-trained on multilingual data work well on cross-lingual reasoning with few-shot settings, is it possible to compare them with some multilingual backbones?
5. I was also wondering whether the performance on MGSM (shown in Table 3) will be affected by adding some text perturbation (here the perturbation should not affect the numeral text), or adding some spans with code-switching.
6. In Table 7, why are the number of few-shot exemplars in different language settings not the same? For instance, only 1 exemplar is provided for ru, th, te, and bn, which is smaller than that of en, de, fr, and es (6 exemplars).


References:
[1] Kojima, Takeshi, Shixiang Shane Gu, Machel Reid, Yutaka Matsuo, and Yusuke Iwasawa. "Large Language Models are Zero-Shot Reasoners." arXiv preprint arXiv:2205.11916 (2022).


**Summary Of The Paper:**

This paper introduced a multilingual arithmetic reasoning task with a corpus named MGSM, and used the few-shot chain-of-thought (Few-Shot CoT) prompting framework to evaluate the large language models’ few-shot learning capability.
Authors empirically found that the GPT-3 and LLM could provide reasonable performance on their constructed test set, and also provided results on two other multilingual reasoning tasks.


**Summary Of The Review:**

This work is interesting and well-motivated, the technical contribution can be extended by tackling the points of questions.

---

> ### Author Response · Authors · 2022-11-17
> **Response to Reviewer 3ofN (Part I)**
>
> We thank the reviewer for the critical comments and valuable suggestions. We were delighted to see that the reviewer found our results to be interesting and our paper to be well-written! In what follows, we would like to provide further clarifications about our work based on their questions and comments:
>
> ### On the Size and Quality of the MGSM Dataset
> One major question I have in mind is, is the current sample size of MGSM sufficient as a benchmark corpus? It would be better if the test sample size can be larger than the current 250, as the original GSM8K contains 1,000 test samples.
> We agree with the reviewer that it would be desirable to have more examples for each underrepresented language in our benchmark. However, due to financial constraints, we could translate only 250 examples from GSM8K to ten typologically different languages by hiring professional translators.
>
> To ensure high-quality, we hired professional translators to perform the translation task. Crowd-sourcing was considered in the early development stages of the project; however, we found that (i) it is actually difficult to find good translators in crowd-sourcing platforms such as Mechanical Turk who were proficient in not only their native (and our target) underrepresented language but also English; (ii) translations by crowd-sourced individuals might not be as high-quality as translations by professional translators. And we note that a similar consideration was also mentioned in (Ponti et al., 2020; [1]) for XCOPA, for instance.
>
> That said, we are trying to garner more funds to extend the scope and size of the MGSM benchmark and hope to update the current dataset with a larger one in the future.
>
> Additionally, as we mention in our response to Reviewer HXqf, there is a strong correlation between performance on 250 examples and the full GSM8K dataset: We evaluate the performance of both LLM-540B and code-davinci-002  with three-shot exemplar chain-of-thought and  direct answer prediction on the English GSM8K test set. We use three sets of different three-shot examples in the prompt, resulting in 12 experiments in total. Across the experiments, the Pearson’s correlation and the Spearman rank correlation coefficient between the performance on the first 250 examples (which correspond to those in MGSM) and the full GSM8K test set as follows: Pearson’s rank correlation is 0.979 and Spearman’s correlation coefficient is 1.0. Such high correlations show that the performance on the first 250 examples is a good predictor of that on the full test set. This suggests that the current setup we have is perhaps sufficient to test the multilingual arithmetic capabilities of language models.
>
> [1] https://aclanthology.org/2020.emnlp-main.185.pdf
>
>
> ### Emergence of Cross-Lingual Reasoning in Large Language Models
> It would be better if authors can add some insight or feature-based analysis on the emergent ability of cross-lingual reasoning of the tested language models GPT-3 and LLM.
> This is an excellent suggestion. In Figure 4, we present the MGSM accuracy scores of GPT-3 and LLM (proprietary model family) with different model scales. It might be useful to provide a detailed discussion of the emergence phenomenon that we observe in these plots. For instance, we see that the model performance (as measured by accuracy) changes significantly as we move from D-1 to D-2 in the GPT-3 model family. It is, however, difficult to draw any concrete conclusions from the LLM family. We will make sure to discuss these observations in more detail in the main context of the paper in the next iteration of our work.
>
> ### Zero-Shot Chain-of-Thought Performance on MGSM
> In Section 4, it would be better if the authors add the experimental results of adopting Zero-Shot CoT (Kojima, Takeshi, et al. 2022) [1], compared with the Few-Shot CoT model.
> We thank the reviewer for this valuable point. We now include the zero-shot chain-of-thought results on LLM-540B, as per the suggestion. The results are as follows:
> |                                     |  avg |  hrl |  lrl |  en  |  de  |  fr  |  es  |  ru  |  zh  |  ja  |  th  |  te  |  bn  |  sw  |
> |-------------------------------------|:----:|:----:|:----:|:----:|:----:|:----:|:----:|:----:|:----:|:----:|:----:|:----:|:----:|:----:|
> | text-davinci-002 (EN-CoT-0shot)     | 29.5 | 37.2 | 13.7 | 46.8 | 40.0 | 36.0 | 42.4 | 40.0 | 33.2 | 31.6 | 19.6 |  6.8 | 15.2 | 13.2 |
> | text-davinci-002 (Native-CoT-0shot) |  9.8 |  7.8 |  3.6 | 46.8 |  8.4 |  4.8 | 13.2 |  6.4 |  5.2 |  8.8 |  5.2 |  1.2 |  2.4 |  5.6 |
> | LLM-540B (EN-CoT-0shot)             | 30.8 | 38.3 | 15.2 | 48.0 | 38.4 | 36.0 | 42.4 | 42.0 | 35.6 | 35.2 | 20.0 | 10.4 | 14.0 | 16.4 |
> | LLM-540B (Native-CoT-0shot)         | 14.4 | 13.2 |  7.7 | 48.0 | 12.8 | 12.4 | 16.8 | 13.6 | 10.8 | 12.8 |  7.6 |  6.8 |  6.8 |  9.6 |
>
> We've also had this updated in our manuscript.

---

> ### Author Response · Authors · 2022-11-17
> **Response to Reviewer 3ofN (Part II)**
>
> > It is a bit surprising that the language models (GPT-3 and LLM) that are not specially pre-trained on multilingual data work well on cross-lingual reasoning with few-shot settings, is it possible to compare them with some multilingual backbones?
>
> We agree that this is surprising; on the other hand, these models are trained with web corpora and without excluding multilingual tokens, so it’s fair to expect these models to have some multilingual ability. As of our submission, we did not find a large-enough model, which is of similar sizes and is trained on multilingual corpora, to compare with; and there is no evidence on the few-shot reasoning ability of smaller models such as multilingual BERT [2] and XLM-Roberta [3].
>
> [2] https://aclanthology.org/N19-1423.pdf
>
> [3] https://aclanthology.org/2020.acl-main.747.pdf
>
> ### Adversarial Perturbations
> > I was also wondering whether the performance on MGSM (shown in Table 3) will be affected by adding some text perturbation (here the perturbation should not affect the numeral text), or adding some spans with code-switching.
>
> Based on our experiments, we found that both GPT-3 (text-davinci-002) and LLM-540B are relatively robust: Neither gets seriously affected by small textual perturbations to the prompt (e.g., insertion of a distractor sentence), so long as the modifications are minor and not applied to the numeral text. However, if the QA format of the prompt gets changed, we then observe noticeable drops in accuracy scores.
>
> ### Regarding Table 7: The Number of Few-Shot Exemplars
> > In Table 7, why are the number of few-shot exemplars in different language settings not the same? For instance, only 1 exemplar is provided for ru, th, te, and bn, which is smaller than that of en, de, fr, and es (6 exemplars).
>
> We apologize for the confusion. As we note in the caption of Table 3, “We use 6 questions and solutions as the few-shot exemplar whenever possible: while the token number for 6-shot prompts in some languages may exceed the token number limit of GPT-3, we use the maximum possible number of exemplars instead for these cases.”
> The number of few-shot exemplars provided for each language is different, mainly because of the tokenization issues of these language models. Most words in English, German, French, and Spanish can be expressed using a few tokens under the tokenizers of these language models; however, the same statement cannot be said for Russian, Thai, Telugu, and Bengali. Sometimes one uses more than eight or ten tokes to express one simple word in these languages using these models’ tokenizers. Due to these tokenization concerns, we could provide only one exemplar for Russian, Thai, Telugu, and Bengali and six examples for English, German, French, and Spanish. Please also feel free to check out our Appendix A.1 for more details and clarifications on this.

---

### Official Review · Reviewer_oVpU · 2022-11-03

**Confidence:** 4
**Correctness:** 3
**Technical Novelty And Significance:** 2
**Empirical Novelty And Significance:** 2
**Recommendation:** 6

**Clarity, Quality, Novelty And Reproducibility:**

This paper is well written and the experiment is well designed. It considers CoT reasoning in a multilingual setup and introduces a new dataset. However, portions of the experiment is impossible to replicate due to the proprietary nature of LLM.

- Is the dataset and CoT explanation publicly available?
- Caption of table 6 has [CITE]?


**Strength And Weaknesses:**

Strength
It introduces a new dataset and presents extensive experiments.

Weakness
- No variance is presented in the few-shot experiment.


**Summary Of The Paper:**

This paper considers chain-of-thought (CoT) in multilingual setup and shows that despite a highly imbalanced pretraining dataset, LLM still learns strong CoT capability in non-English. It extends an existing dataset to multiple languages. It also compares different ways to construct exemplar and CoT.


**Summary Of The Review:**

This paper considers CoT in multilingual setup and creates a new dataset, despite minor issues of the experiments.

---

> ### Author Response · Authors · 2022-11-17
> **Response to Reviewer oVpU**
>
> We thank the reviewer for the comments and feedback. We are delighted to see that the reviewer has found our paper to be well written and our experiments to be well designed. In what follows, we would like to provide further clarifications about our work based on their questions and comments:
>
> ### Statistical Significance
> We agree with the reviewer that it would be desirable to run our experiments multiple times and estimate statistical significance and confidence bounds, but due to computational constraints, we were not able to run enough experiments to calculate reliable statistical significance. That said, we found, through our preliminary investigations in the early stages of this work, that our results were reliable and robust, and the magnitude and consistency of the gains over the baselines across multiple languages confirm this belief.
>
> ### MGSM Dataset and CoT Prompts
> Yes, we included both our proposed MGSM corpus and the CoT prompts used in our experiments in the Supplementary Materials of our submission. We will also release all the data, code, and Codex outputs upon the publication of our work.
>
> ### Proprietary Nature of LLM
> We are sorry that we are not allowed to share the outputs of the LLM model at this time, due to its proprietary nature. However, the users may reproduce our results on the publicly available datasets with OpenAI models such as text-davinci-002 and code-davinci-002.
>
> ### Additional Comments and Questions
> > Caption of table 6 has [CITE]?
>
> We thank the reviewer for catching this minor error. We have fixed that in our updated draft.

---

### Official Review · Reviewer_HXqf · 2022-11-04

**Confidence:** 4
**Correctness:** 4
**Technical Novelty And Significance:** 2
**Empirical Novelty And Significance:** 2
**Recommendation:** 6

**Clarity, Quality, Novelty And Reproducibility:**

**Clarification**
* In the "Manual translation process" passage, who are the popular MT providers?

**Questions to the authors**
* Did you check the math background of translators?
* How do you measure the quality of translated examples compared to English examples?
* How does the pre-training (i.e., datasets, model design) of LLMs affect their multilingual abilities?

**Details Of Ethics Concerns:**

There is no ethical issue.

**Strength And Weaknesses:**

**Strengths**
* The multilingual benchmark is translated by professional translators and verified by an additional one for a random subset.
* Experiments are well-designed with 4 prompting methods and 3 types of input exemplars.
* Intensive analysis provides new insights for multilingual reasoning abilities of large language models.
* The paper is also well-written.

**Weaknesses**
* The number of examples is not significant, with only 250 examples, compared to GMS8K with 8.5K examples, so the results may not be generalized if more examples are translated and added to the benchmark.
* The verification process is less discussed, and not convincing to guarantee the quality of the benchmark.
* The approach of translating a problem to English does not test the multilingual reasoning abilities of LLMs.

**Summary Of The Paper:**

The paper introduces a benchmark containing 250 grade-school math problems translated from English to ten languages and also provides a deep analysis to explore multilingual reasoning abilities of two SotA large language models GPT-3 and LLM-540B in terms of both chain-of-thought and commonsense reasoning. The proposed benchmark was translated by native speakers of the target language with at least two years of professional experience. The experiments are well-designed, and the results are enough to support the claims. They find that on the MGSM benchmark, LLM-540B shows superior multilingual reasoning ability compared to GPT-3 on all languages with different settings. Similarly, for the commonsense reasoning benchmark, LLM-540B also outperforms the SotA models: RoBERTa Large and XLM-R Large on two other benchmarks: XCOPA and XL-WiC, respectively.

**Summary Of The Review:**

Although the findings are interesting, the contributions are not significant enough for the multilingual research since the benchmark is small and the verification process is not well described (or implemented?), which may flip the current conclusions/claims if more multilingual examples with different types of problems are added. The big plus of this paper is the intensive experiments and detailed analysis.

---

> ### Author Response · Authors · 2022-11-17
> **Response to Reviewer HXqf**
>
> We thank the reviewer for the critical comments and valuable suggestions! We are delighted to see that the reviewer has found our work to be well-presented and our results to be well-supported. In what follows, we would like to provide further clarifications about our work based on their questions and comments:
>
> ### Size and Quality of the Benchmark
> We agree with the reviewer that it would be desirable to have more examples for each language, but due to financial constraints, we were able to translate only 250 examples from GSM8K to ten typologically different languages with the help of professional translators. We initially considered crowd-sourcing to contain more samples and cover more languages; however, we were afraid that crowd-sourcing might result in lower-quality translations. (In fact, Ponti et al. (2020) also mention a similar concern for their XCOPA work [1].) We note that many other cross-lingual/multilingual benchmarks, such as XCOPA, are of the same or similar size. That said, we have been attempting to garner more funds to extend the scope and size of the MGSM benchmark and hope to update the current dataset with a larger one in the future.
>
> Additionally, we would like to mention the following correlation between performance on 250 examples and the full GSM8K dataset: We evaluate the performance of both LLM-540B and code-davinci-002  with  3-shot exemplar chain-of-thought and  direct answer prediction on the English GSM8K test set. We use 3 sets of different 3-shot examples in the prompt, resulting in 12 experiments in total. Across the experiments, the Pearson’s correlation and the Spearman rank correlation coefficient between the performance on the first 250 examples (which correspond to those in MGSM) and the full GSM8K test set as follows: Pearson’s rank correlation is 0.979 and Spearman’s correlation coefficient is 1.0. Such high correlations show that the performance on the first 250 examples is a good predictor of that on the full test set.
>
> [1] https://aclanthology.org/2020.emnlp-main.185.pdf
>
>
> ### Responses to Additional Questions and Comments:
>
> > In the "Manual translation process" passage, who are the popular MT providers?
>
> We used the professional translation service provided by a translation service company (name anonymized to comply the double-blind policy, and we will deanonymize it upon publication). This process is done internally within their service and we unfortunately don’t have an exhaustive list of their MT translations to compare with, however, the list includes Google Translate and Bing Microsoft Translator. We have had this clarified in our draft.
>
> > How do you measure the quality of translated examples compared to English examples?
>
> As mentioned above, the verification is done internally by the translation service company. Examples in both English and the target language are sent to a second translator to double check the faithfulness. We  have had this clarified in our draft.
>
> > How does the pre-training (i.e., datasets, model design) of LLMs affect their multilingual abilities?
>
> Given the size of the model and computational resources, we are not able to train many models with careful control on datasets and model designs due to the resource cost. We leave the answer to this question for future work for either us or others.

---

> > ### Comment · Reviewer_HXqf · 2022-12-12
> > **Benchmark Quality Concerns**
> >
> > Thank you very much for your time addressing my questions and concerns!
> >
> > In terms of benchmark quality, it is still unclear to some extent since it was done internally by a third party. Also, what happens if a second translator does not agree with the translation? It would be great if you can elaborate on both annotation and verification processes to guarantee high quality for the proposed benchmark given the small scale. However, I believe that the paper has some merits and an interesting analysis plus the comparison between two of the best LLMs in terms of multilingual abilities, so I raised the score to 6.

---

### Author Response · Authors · 2022-11-17
**General Response to the Reviewers**

We thank all the six reviewers for their valuable feedback and comprehensive suggestions.

To summarize, our work (i) introduces  a new multilingual benchmark for arithmetic reasoning called MGSM, (ii) conducts extensive experiments on both MGSM and other multilingual reasoning benchmarks (such as XCOPA and WiC) to study the multilingual reasoning capabilities of large language models, and (iii) illustrates that when equipped with multilingual chain-of-thought prompts, these language models can perform on par with, and even sometimes outperform, supervised (finetuned) multilingual models. To the best of our knowledge, this is the first study to demonstrate that large-scale language models can perform multilingual chain-of-thought reasoning.

We are delighted to see that all the reviewers agree that our work is well-written and that it offers interesting insights. Reviewers [oVpU], [3ofN, [q52S], and [DTUx] also mentioned that our empirical results were well-motivated and well-supported. There were, however, some reasonable concerns by Reviewers [HXqf] and [3ofN] about the limited size and quality of the MGSM benchmark. We agree with them that it would be indeed desirable to have a larger and more comprehensive multilingual benchmark for arithmetic reasoning that would not only contain more than 250 examples for each target language but also include even more low-resource languages. However, due to financial constraints, we could only include 250 examples for each of the ten typologically diverse languages. We also want to note that to ensure high-quality, we hired professional translators to help us with the translations. We are now trying to garner more funds to be able to extend our work and hope that in the future, we will be able to extend the scope and size of our benchmark.

On the topic of reproducibility, Reviewers [oVpU], [aQqu], and [DTUx] expressed their concerns about replicating the results of our work. As we also mention in our individual responses to the reviewers, we would like to share that we have already provided the full set of the direct and chain-of-thought prompts used in our experiments in our original submission (please refer to the supplementary materials). And we will release our current supplementary materials, as well as the Codex and GPT-3 model outputs,  upon the publication of our work so that researchers can make meaningful comparisons with our results.

A few reviewers such as Reviewer [3ofN] asked clarifying questions about our results. We attempted to answer them in both our revised draft and our individual response. And if the reviewers have any further clarifying questions, we would be delighted to answer them to the best of our abilities.

Finally, Reviewer [3ofN] asked about the different number of few-shot examples used in some of our multilingual experiments. We answered the reviewer’s question in our individual response to them, but we also would like to clarify this point for the other reviews to address any potential confusion. The number of few-shot exemplars provided for each language is different, mainly because of the tokenization issues of the language models used in our experiments. Most words in English, German, French, and Spanish can be expressed using a few tokens under the tokenizers of these language models; however, the same statement cannot be said for Russian, Thai, Telugu, and Bengali. Sometimes one might use more than eight or ten tokes to express one word in these languages using these models’ tokenizers. Due to these tokenization concerns, we could provide only one exemplar for Russian, Thai, Telugu, and Bengali and six examples for English, German, French, and Spanish. We refer the reviewers to checkout our Section A.1 in the Appendix for more details and clarifications on this topic.

We thank all the reviewers and the AC for the reviews. If there is anything we might have missed, or if there are any further questions about the paper, please let us know and we are more than happy to work on it.

---

### Decision · Program_Chairs · 2023-01-20

**Decision:**

Accept: poster

**Justification For Why Not Higher Score:**

Nice paper but with some limitations.

**Justification For Why Not Lower Score:**

The influential nature of the conclusions makes the paper a valuable contribution for the community.

**Metareview: Summary, Strengths And Weaknesses:**

The paper introduces the first multilingual benchmark for evaluating the task of arithmetic reasoning. It is built from the translation by experienced professionals of part of an English dataset. The objective is to evaluate the multilingual abilities of very large language models trained on monolingual English corpora incorporating some elements of other languages as well. The authors provide an in-depth analysis of two large language models for reasoning tasks (multilingual arithmetic and common-sense reasoning). It is found that by using recent prompting strategies, these large models perform on par or better than models fine-tuned on a target language and outperform SOTA models.

The main contributions of the paper are (i) the introduction of a multilingual dataset for evaluating language models reasoning when the existing benchmarks are in English, (ii) an in depth analysis of the reasoning abilities of language models. Their conclusion is rather surprising: when they reach a critical size, large English monolingual models do perform extremely well on multilingual reasoning, sometimes better than language specific trained models.
All the reviewers agree that this contribution could greatly influence future work. The paper has also some limitations: the size of the translated corpus is small and the technical contribution itself is not so original as the experiments follow the recent prompting practice. However, the added value of the work lies in the conclusions made possible by the corpus and demonstrated in the analysis. During the discussion, the reviewers highlighted the potential influence of this work on future practices: this could modify the nature of research on multilingual models.


**Note From Pc:**

if the above contains the word "oral" or "spotlight" please see: "oral" presentation means -> notable-top-5% and "spotlight" means -> notable-top-25%. As stated in our emails, we are disassociating presentation type from AC recommendations

**Summary Of Ac-Reviewer Meeting:**

We had a meeting with two reviewers HXqf and q525. They both agree that the paper has some drawbacks mainly related to the small size of the corpus – translating in 10 languages with professional translators has a cost. They however think that the paper reveals a surprising new phenomenon that could really be influential and that the community could largely benefit from this finding.